# An Automotive Reference Testbed with Trusted Security Services

Teri Lenard [1,2,*], Béla Genge [2], Piroska Haller [2], Anastasija Collen [1] and Niels Alexander Nijdam [1]

1   Centre Universitaire d'Informatique, Geneva School of Economics and Management, University of Geneva, Route de Drize 7, CH-1227 Carouge, Switzerland
2   Faculty of Engineering and Information Technology, George Emil Palade University of Medicine, Pharmacy, Science, and Technology, 38 Gheorghe Marinescu Street, 540142 Targu Mures, Romania
*   Correspondence: teri.lenard@unige.ch

**Abstract:** While research in the field of automotive systems inclined in the past years towards technologies such as Vehicle-to-Everything (V2X) or Connected and Automated Vehicle (CAV), the underlying system security still plays a crucial role in assuring trust and system safety. The work at hand tackles the issue of automotive system security by designing a multi-service security system specially tailored for in-vehicle networks. The proposed trusted security services leverage Trusted Platform Module (TPM) to store secrets and manage and exchange cryptographic keys. To showcase how security services can be implemented in a in-vehicle network, a Reference TestBed (RTB) was developed. In the RTB, encryption and authentication keys are periodically exchanged, data is sent authenticated, the network is monitored by a Stateful Firewall and Intrusion Detection System (SF/IDS), and security events are logged and reported. A formal individual and multi-protocol analysis was conducted to demonstrated the feasibility of the proposed services from a theoretical point of view. Two distinct scenarios were considered to present the workflow and interaction between the proposed services. Lastly, performance measurements on the reference hardware are provided.

**Keywords:** automotive; controller area network; security services; trusted platform module; testbed





## 1. Introduction

Current research in the automotive field leans towards interconnected and cooperative vehicles, with Vehicle-to-Everything (V2X) [1] being the new *hot-topic*, only to be accompanied by Connected and Automated Vehicles (CAVs) [2]. From time to time, we have to take a step back and reconsider the foundation on top of which these promising technologies are built. The sub-systems and communication protocols present in our everyday vehicles were designed with a clear scope and purpose, with the focus not on security features, but rather on *mission-critical* aspects, such as real-time message transmission, network reliability, and tolerance to errors. Modern vehicles incorporate multiple communication systems to satisfy the wide variety of functionalities required by the vehicles and their users. A frequently used protocol for frame exchange between Electronic Control Units (ECUs), sensors, and sub-networks, is the Controller Area Network (CAN) [3]. Vehicles contain communication protocols for computationally restricted sensors (e.g, SENT [4]), infotainment systems (e.g., MOST [5]), or for communication with the surrounding infrastructure (e.g., IEEE 802.11p [6]).

Like any other networking systems, the security of vehicle networks can be compromised. ECUs responsible for the automation and control of vital vehicle functionalities can be re-flashed to execute malicious firmware [7]. Infotainment systems can be compromised, and as a consequence malicious actors become capable of controlling the vehicle itself [8]. Similarly, the mileage of odometers can be tampered with [9], and sensors can be manipulated to enable and disable after-treatment systems [10]. Although these statements paint a grim scene, academic researchers, standardization units, and the industry have joined forces to overcome the mentioned issues. Academic efforts bring forward new

and improved security solutions every year (e.g., data authentication, key distribution protocols) [11], sensor data privacy preserving techniques [12], and advanced anomaly detection [10]. Industry and standardization units have established security standards and recommendations for manufacturers (e.g, ISO 21434, UNECE R155, and R156). Further recommendations were brought up by Automotive Open System Architecture (AUTOSAR)'s Secure Onboard Communication (SecOC) [13] standard, or by the Trusted Computing Group (TCG)'s guidelines to Trusted Platform Module (TPM) [14] usage in automotive vehicles [15]. Security problems are born when a system is not designed with security in mind from the beginning, and when they do not follow already-known best practices in protocol design [16]. To pave the roadmap of academic security research integration into real automotive environments, we identified several concerns present in our past works and the related literature:

i    While proposing novel and improved security services may represent a significant contribution to the literature, in the real world, most systems are composed of multiple protocols concurrently running over the same communication medium, which can raise several security concerns if not designed and implemented properly [17].

ii    In the most frequently used communication protocol for vehicle networks, the CAN protocol, there is a gap in terms of identifying the source and destination of a given message due to its bus-oriented communication pattern. This aspect becomes a severe concern when security comes into discussion. Preferably, each actor, (e.g., the security service) should be capable of identifying message sources and corresponding services.

iii    A distinct aspect of vehicular networks is that typically, they are not attacked from outside (e.g., via the Internet), but from inside, by actors who have direct physical access to the system with or without the owner's consent (e.g., engine tuning, installation of custom firmware and devices). Consequently, the problem of where to store cryptographic credentials becomes critical.

iv    Experimentation represents a key aspect in validating the theoretical protocol design after a proper formal verification. For this reason, building and reproducing a reference architecture for testing purposes is a must.

The paper at hand addresses each of the aforementioned concerns with three distinct contributions as follows:

- *Contribution I*: Several issues were identified in the previously proposed security services [18,19] regarding protocol and message identification, authentication, aliveness, and agreement between protocol participants. We address those problems by improving the protocols in terms of security design and best practices [16]. To better understand the security implications, a threat model and evaluation is presented. Furthermore, a three-step formal security analysis was conducted comprising both individual and multi-service analysis.
- *Contribution II*: An approach is formulated to describe how TPMs can be used in automotive networks. This covers multi-service identification, secure key storage, key distribution, and log attestation. Performance measurements of TPM commands are offered on the considered hardware.
- *Contribution III*: A system of trusted security services is proposed designed for CAN. An affordable and reproducible Reference TestBed (RTB) was developed to showcase the implementation of automotive security services. In the RTB, data is sent authenticated using the Mixed Data Authentication for CAN (MixCAN) protocol [18], long- and short-term cryptographic keys are periodically exchanged [19], the network traffic is monitored via a Stateful Firewall (SF) and an Intrusion Detection System (IDS) [20], and finally, security events are logged and published to trusted services.

The rest of the paper continues with the related work and a background on vehicle networks and TPM functionalities in Section 2. In Section 3, the improvements considered when designing the security protocols are outlined. In Section 4, our proposed system and security services are described. Following that, the security analysis in Section 5 is

conducted. Subsequently, the reference testbed is described in Section 6. The paper offers a series of discussions in Section 7, and plans for future work in Section 8. The paper concludes in Section 9.

## 2. Background and Related Work

### 2.1. Vehicle Networks

Vehicle networks incorporate several different communication protocols. One of the most commonly used is the Controller Area Network (CAN) protocol, initially developed by Robert Bosch GmbH, and later standardized by the International Standardization Organization [3]. CAN is a bus-oriented protocol that allows different control units and sensors to exchange frames under a reliable, error-prone medium. Along the CAN protocol, a vehicle network may contain communication protocols such as Local Interconnected Network (LIN) for offering different services to the vehicle passengers (e.g., door/seat control, climate system), Single Edge Nibble Transmission (SENT) for resource-restricted digital sensors, or Media-oriented System Transport (MOST) for media-oriented control units and services.

There are several standardized versions of the CAN protocol, such as the Controller Area Network with Flexible Data-rate (CAN-FD) [21], developed by Robert Bosch GmbH to improve the original limitations in terms of bandwidth, transfer rates, or the amount of data that can be transmitted in one frame. Other high-level protocols built over CAN are SAE J1939 for in-vehicle communication of heavy-duty vehicles, ISO-TP and UDS for automotive diagnostics, and CANopen for rail vehicles. While CAN allows frame recipients to acknowledge and detect errors on the frames received, the standard is not concerned with secure communication. For this, standardization units such as AUTOSAR are responsible for building specific guidelines.

The article focuses on CAN systems. In CAN, a frame is broadcast over a common bus. A CAN frame is identified by a 11-bit identifier field, contains 8 bytes of data, and in addition to other fields, leverages a 15-bit error correction checksum (e.g., a cyclic redundancy check). In protocols such as CAN-FD, the data field is expanded with 64 bytes of data. In other protocol variations (e.g., SAE J1939), functionalities such as frame addressing are extended through a 29-bit identifier, and with an address-claiming procedure.

A simplified view of the mentioned protocols is shown in Figure 1. By analyzing the communication between different ECUs, including the connectivity between the CAN network to the gateway and to the Communication Control Unit (CCU), we propose a multi-service security system implemented in a reference architecture that replicates a real CAN system. Consequently, a system which was initially insecure was extended with the necessary security services to achieve a higher level of resilience against internal and external threats.

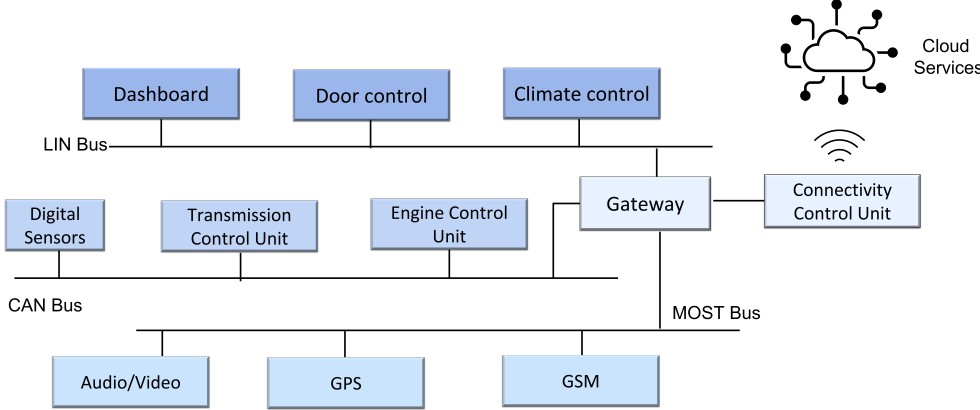

**Figure 1.** Overview of the components and communication channels frequently found inside automotive systems.

### 2.2. Trusted Platform Module

Designed and developed by the Trusted Computing Group (TCG) [15], the Trusted Platform Module (TPM) [14] standard represents a tamper-resistant cryptographic co-processor, capable of executing cryptographic operations securely and separately from the main processing unit. While the initial development of the TPM standard targeted traditional information systems, recently the TCG published an automotive related set of recommendations that describes specially tailored profiles for automotive networks [15]. Additionally, the TCG formed the Vehicle Services Working Group (https://trustedcomputinggroup.org/work-groups/vehicle-services/, accessed on 16 January 2023), intending to address problems related to adopting trust-related standards (e.g., TPM) into automotive environments.

As such, the main scope of the TPM standard is to empower control units (e.g., ECUs or CCUs), with standardized, state-of-the-art cryptographic algorithms and operations to obtain a more secure and trusted environment. The functionalities of the TPM range from secure key generation processes, using internal key hierarchies [22], to digital signature computation and verification, hashing, encryption, or random number generation. In addition to these fundamental features, TPMs offer additional secure key storage, and Platform Configuration Register (PCR) banks targeted for remote attestation.

There are two main options for storing secret keys using the TPM. The first option is to store the key persistently in the TPM's non-volatile memory. The drawback here is in the limited amount of non-volatile memory, and as well as key storage, the non-volatile memory may need to serve different storage scopes (e.g., monotonic counters, secret storage). The second option is more effective and uses a special key inside the TPM called Storage Root Key (SRK). The SRK is rooted into the TPM, meaning that it is only accessible internally and it can never leave the TPM. SRK is created when a TPM user takes ownership of the TPM, and its main purpose is to protect other applications' keys. Using an operation called *sealing*, the private part of a pair of keys is encrypted with the SRK, while the public part remains available. By doing so, the key becomes available only to the TPM itself. Another type of special key offered by the TPM is the Endorsement Key (EK). This is a permanent key embedded in the TPM by the Original Equipment Manufacturer (OEM). The private portion remains stored in the TPM, while the public part is available to the outside word. This key mainly serves as a proof to identify genuine TPMs.

Lastly, an additional security feature is the PCR banks [23], which ultimately are a set of memory locations with unique properties. PCRs utilize a cryptographic hash function to compute measurements that can be used to attest the integrity of log messages or other forms of events. Basically, PCRs support a single operation that *extends* the current value of the PCR with a hash value provided. In other words, it concatenates the current PCR value with a new digest value, and stores the newly obtained result after hashing.

### 2.3. Related Work

The domain of security services in the context of automotive systems represents a research field thoroughly addressed in the past year by individual works and review papers. Martínez-Cruz et al. [24] offer a comprehensive survey of issues, threats, challenges, and the most relevant solutions targeting the security of in-vehicle systems. Similarly, Lokman et al. [25] cover the topic of IDS in automotive systems by providing a comparative analysis of existing IDSs. More recent works, such as Rathore et al. [7], motivate further the need for solid security mechanisms. The authors mention their concern regarding the continuous evolution of vehicle networks (e.g., smart and interconnected vehicles), and the threats that may arise as a consequence of this process.

To further address future security concerns in automotive systems, the work of Pham and Xiong [26] explores security threats and existing solutions targeting CAV specifically. Since automotive systems become more and more connected, the technology is pushed towards inter-vehicle communication and persistent connection with external services, therefore it is difficult to estimate what impact threats will have on system security [27].

It is evident that a baseline of works such as [25,26,28] exist which can serve as a starting point to build reliable, secure, and safe automotive systems. Since the threats, counter measures, attacks vectors, and solutions are known [29], the pathway of securing vehicle networks can be paved. The additional contribution that the work at hand provides relates to the proposed trusted multi-service system. It intends to demonstrate how an in-vehicle network can be secure by respecting security concerns identified in the related works.

## 3. Design Considerations

Previously proposed protocols [18,19] lacked security properties (such as aliveness, secrecy, or agreement) which are addressed in the improved versions and in the security analysis from Section 5. Consequently, the following design improvements are defined below and later addressed in accordance to Ferguson et al. [16]. Further notations used can be viewed in Table 1.

**Table 1.** Table of symbols leveraged in the formal protocol description.

| Symbol | Description |
|---|---|
| $i$ | Protocol initiator |
| $r$ | Protocol responder |
| $a$ | Protocol adversary/attacker |
| $d$ | Key distributor |
| $m$ | Protocol member |
| $M$ | Set of protocol members |
| $p(i,r)$ | Protocol between $i$ and $r$ |
| $p(r), p(i)$ | Protocol executed by $i$ or $r$ |
| $k(i,r)$ | Symmetric short-term key between $i$ and $r$ |
| $K(i,r)$ | Symmetric long-term key between $i$ and $r$ |
| $pk(i)$ | Public key of $i$ |
| $sk(i)$ | Secret key of $i$ |
| $n$ | Freshness nonce |
| $p_{id}$ | Protocol identifier |
| $k_{id}$ | Key identifier |
| $m_{id}$ | Message identifier |
| $pm_{id}$ | Private message identifier |
| $mc$ | Monotonic counter |
| $\{\}$ | Encryption/sealing |
| $\|$ | Concatenation |
| $\{\}_{k(i,r)}$ | Symmetric encryption with $k(i,r)$ |
| $\{\}_{pk(i)}$ | Asymmetric encryption with public key $pk(i)$ |
| $\{\}_{sk(i)}$ | Asymmetric encryption/digital signature with private key $sk(i)$ |
| $p_{type}$ | Protocol type (LTK or STK) |
| $f_r, f_s$ | Frames received and sent |
| $E_{bf}(m)$ | Encrypted bloom filter function over message $m$ |

### 3.1. Service Identity

TPMs offer a convenient mechanism named sealing, through which a key can be linked to a specific TPM. The private part becomes protected by the TPM and only usable within the TPM, while the public part remains available and can be shared with other trusted parties. This process is beneficial for multi-protocol systems, since it allows a system member (e.g., service) to identify a specific service being executed with a specific TPM. This further implies

that the service messages originate from a trusted agent (e.g., service). In other words, by having a Service Identity (SI), multiple services of the same kind can be executed on a single piece of equipment. Consequently, each service can be uniquely identified and associated with the equipment and TPM. This process is outlined in the system bootstrapping process.

Following this rational, the definition below is given:

**Definition 1.** *A Service Identity (SI) represents the property obtained by a service as an outcome of linking its cryptographic key to a TPM. Consequently, the SI allows service identification at service and hardware level.*

### 3.2. Identification, Numbering, and Freshness

A crucial aspect that must be taken into consideration while designing security protocols is clear message identification and freshness. In CAN networks, each frame has an associated predefined identifier field. This identification mechanism is only meant to refer to the message itself, not to the data it contains, nor to the message source (e.g., service). In addition to this, each protocol proposed in the current work leverages three additional identification and freshness terms: Protocol Identifier (PID), Message Identifier (MID), and Private Message Number (PMN).

**Definition 2.** *A Protocol Identifier (PID) represents a constant value across the system, meant to identify the protocol and the protocol version executed by a service.*

PIDs are an essential factor in multi-protocol environments. From a security perspective, each PIDs $p_{id}$ is not secret, and is distributed without preserving its confidentiality. Consequently, services should ensure only its integrity. Depending on the protocol implementation, the $p_{id}$ can be sent once when the protocol is initiated, or with every message.

**Definition 3.** *A Message Identifier (MID) represents a monotonic counter value used to identify, count, and re-assemble messages in a protocol.*

The identifier $m_{id}$ is meant to help in message counting, reconstructions, and error detection. Similarly, this is a publicly transmitted term for which only the integrity is maintained.

**Definition 4.** *A Private Message Number (PMN) is the secret equivalent of MID. A PMN $pm_{id}$ is meant to identify messages, and serves as an additional source of freshness that hides information about a nonce or freshness monotonic counter mc.*

In automotive systems, message freshness can be ensured in several manners, ranging from monotonic counters and timestamps to lifetime freshness values. The proposed protocol leverages pre-installed monotonic counters $mc$ stored in the non-volatile memory of the TPM. Additionally, a private message numbering $pm_{id}$ is introduced as a secondary source of freshness and private identification. Each protocol ensures for a freshness $n$ that is the concatenated result of this two values $n = (pm_{id} \| mc)$.

### 3.3. Security Properties

Security properties are deemed to be necessary in security protocols for correctness. Cremers [30] considered that security properties represent an essential part of the security protocol design, defining guarantees in terms of security, which must be met by the protocol design. According to [30], the following security properties were identified:

- *Secrecy.* The secrecy property guarantees that certain information in a security protocol is not revealed to an attacker, even if the protocol is executed over an untrusted communication channel.

- *Authentication.* Intends to guarantee a protocol participant that there exists a second credible and trustworthy participant with which it communicates and executes the protocol.
- *Agreement.* The agreement property must guarantee that after completing a protocol run, all protocol participants agree on the messages exchanged.

## 4. Proposed System

The proposed security system, as illustrated in Figure 2, incorporates multiple control units executing our proposed security services. Usually, a CAN system includes multiple ECUs, digital sensors, and a CCU for cloud communication. The proposed architecture follows the same approach.

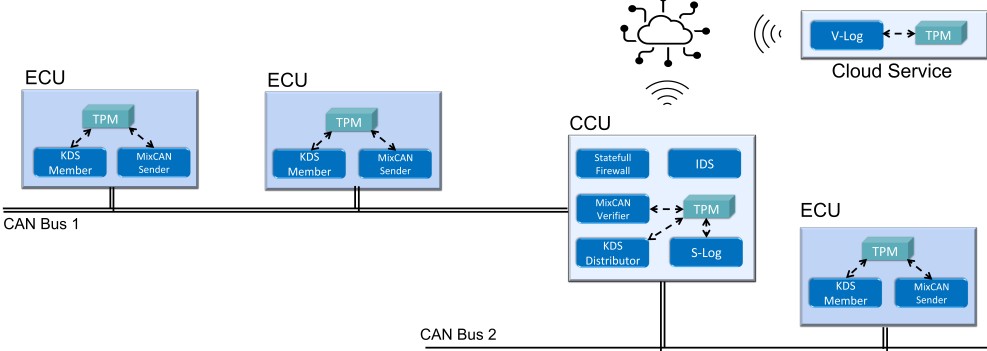

**Figure 2.** Enhanced in-vehicle network architecture with trusted security services and Trusted Platform Module (TPM).

The system envisions that each network member has attached a physical TPM and was bootstrapped accordingly by a trusted authority. This simplified architecture is meant to present the fundamental functionalities of the systems. Due to the design's considerations, in terms of protocol design and implementation, the system can be easily expanded with additional control units, protocol instances (e.g., running two Key Distribution Service (KDS) instances in parallel), and networks (e.g., adding a new CAN connection).

Each ECU is responsible to participate in the KDS as a key receiver, and in the data authentication protocol MixCAN as both a sender and receiver. On the CCU side, the KDS distributor is executed together with a MixCAN sender/receiver, a Stateful Firewall and Intrusion Detection System (SF/IDS), and a Secure Logging (S-Log) instance. S-Log is responsible for communicating via a cloud connection relevant security alerts to Logging Verifier (V-Log) to validate and store the received information. The system contains the following services:

- *Key Distribution Service (KDS)*: distributes long-term encryption keys, and short-term data authentication keys to a group of protocol members by leveraging asymmetric cryptography.
- *Mixed Data Authentication for CAN (MixCAN)*: aggregates and authenticates a mix of frames using Message Authentication Codes (MACs) computed with the KDS short-term authentication key.
- *SF*: monitors sequences of CAN traffic in a stateful manner based on CAN frame identifier field.
- *IDS*: extends the functionalities of the SF by performing additional deep-packet inspection at CAN data frame level, and by monitoring transmission times between frames.
- *S-Log*: generates security alerts, which ultimately are log events, yielded by other security services, signed with the TPM, and chained together for attestation using PCRs.

### 4.1. Bootstrapping

The bootstrapping process is presented with two distinct algorithms, covering the requirements for the KDS and the S-Log service. Both algorithms use function notations from

the TPM2 software stack (TSS) tpm2_tools (https://github.com/tpm2-software/tpm2-tools, accessed on 16 January 2023) to describe the interactions with the TPM. The first bootstrapping procedure is outlined in Algorithm 1. An instance of KDS implies the presence of two roles: a key distributor $d$, and a set of key receiver members $m \in M$. Each $m \in M$ is required to know the long-term public key of $d$, denoted by $pk_d$, while $d$ is required to know the shared long-term key $pk_m$ by each $m \in M$. In the first function, `bootstrapDistributor`, a new key is generated using the TPM, and is stored using sealing, denoted by the operator {}. Afterwards, it is copied over a secure channel to each $m \in M$ and individually loaded into the corresponding TPM. In the second function `bootstrapMember`, the first step consists of generating a new pair of keys (e.g., using openssl). Afterwards, for each $m \in M$, the new pair of keys is securely copied, loaded, and sealed in the TPM.

Algorithm 2 outlines the steps taken in bootstrapping the S-Log and V-Log. The $pk_l$ and $sk_l$ are a pair of asymmetric keys of S-Log. Algorithm 2 contains two functions, `bootstrapLogger()` is first executed on the in-vehicle service, and `bootstrapLogVerifier()` afterwards on the remote V-Log. In both algorithms, the function calls `copy()` and `receive()` refer to transmitting the referred terms to the corresponding address over a secure channel.

---

**Algorithm 1:** Bootstrapping KDS

   **Data:**    $d$: Protocol key distributor;
      $m \in M$: Protocol receiver member;
   **Result:**    $pk_d$: Public key of $d$;
      $sk_d$: Secret key of $d$;
      $pk_m$: Public key of $m$;
      $sk_m$: Secret key of $m$;
   **Function** `bootstrapDistributor`$(d, m \in M)$:
      $pk_d, \{sk_d\} \leftarrow$ `tpm2_create()`
      **For** $m \in M$ :
         `copy(`$pk_d$`, `$m$`)`
      `receive(`$pk_m$`, `$m$`)`
      `tpm2_load(`$pk_m$`)`
      **return** $pk_d, sk_d, pk_m$
   **Function** `bootstrapMember`$(d, m \in M)$:
      $pk_m, sk_m \leftarrow$ `gen_key()`
      `copy(`$pk_m, d$`)`
      **For** $m \in M$ :
         `copy(`$pk_m$`, `$sk_m$`, `$m$`)`
         `tpm2_import(`$pk_m$`, `$sk_m$`)`
         `tpm2_seal(`$sk_m$`)`
         `receive(`$pk_d$`, `$d$`)`
         `tpm2_load(`$pk_d$`)`
      **return** $pk_d, sk_m, pk_m$

---

**Algorithm 2:** Bootstrapping the S-Log and V-Log

   **Result:**    $pk_l$: Public key of S-Log;
      $\{sk_l\}$: Sealed secret key of S-Log;
   **Function** `bootstrapLogger()`:
      $pk_l, \{sk_l\} \leftarrow$ `tpm2_create()`
      `copy(`$pk_l$`, V-Log)`
      **return** $pk_l, \{sk_l\}$
   **Function** `bootstrapLogVerifier()`:
      $pk_l \leftarrow$ `receive(`$pk_l, s\text{-}log$`)`
      `tpm2_load(`$pk_l$`)`
      **return** $pk_l$

*4.2. Key Distribution Service*

In a KDS instance there is a set of *m* key receivers, and a single key distributor *d*. A round of the protocol is started by a protocol initiator *i* in relation to a set of receivers *r*. Initially proposed by Genge and Haller [19], KDS consists of three key distribution processes: Long-term Sey Protocol (Proto-LTK), Short-term Key Protocol (Proto-STK), and symmetric Key Synchronization (SKS). For convenience, only the improved KDS version is described.

In Proto-LTK, the communication initiator and key distributor are denoted by $ECU_i^d$, and a message receiver by $ECU_r^m$. To run a round of Proto-LTK, $ECU_i^d$ broadcasts to each $ECU_r^m$ a sequence of messages consisting of a public part, a private part, and an authentication part. If $ECU_r^m$ manages to verify the sequence received, it responds back with a confirmation and a proof demonstrating the usage of the new key. If an error occurs, $ECU_r^m$ is required to run SKS.

$$ECU_i^d \rightarrow ECU_r^m : m_{id}, p_{id}, \{k_{id}, n, K(i,r)\}_{pk(r)}, \{m_{id}, p_{id}, \{k_{id}, n, K(i,r)\}_{pk(r)}\}_{sk(i)}$$
$$ECU_r^m \rightarrow ECU_i^d : m_{id}+1, p_{id}, \{m_{id}+1, n+1, p_{id}\}_{K(i,r)}. \tag{1}$$

In Equation (1), the communication terms can be viewed. In the first interaction, $ECU_i^d$ sends $m_{id}$ and $p_{id}$ as plain text (e.g., public part), accompanied by the encrypted part with the public key shared by each $ECU_r^m$. The encryption is performed over a unique key identifier $k_{id}$, a freshness *n*, and the new long-term encryption key $K(i,r)$. Lastly, both the public part and the secret one are signed with the $ECU_i^d$ private key. If $ECU_r^m$ successfully verifies the received terms, it responds with a incremented $m_{id}+1$ and freshness $n+1$, the same *pid*, and an encryption action performed with the new key. In this way, $ECU_i^d$ can confirm that the new key was distributed correctly.

Proto-STK depends on Proto-LTK since it leverages the encryption key obtained from it to distribute a short-term authentication key. The roles in Proto-STK are unchanged from Proto-LTK, and the protocol follows a similar message structure.

$$ECU_i^d \rightarrow ECU_r^m : m_{id}, p_{id}, \{k_{id}, n, k(i,r)\}_{K(i,r)}, \{m_{id}, p_{id}, \{k_{id}, n, k(i,r)\}_{K(i,r)}\}_{sk(i)}$$
$$ECU_i^m \rightarrow ECU_r^d : m_{id}+1, p_{id}, \{m_{id}+1, p_{id}, n+1\}_{k(i,r)}. \tag{2}$$

As stated in Equation (2), $ECU_i^d$ broadcasts a sequence of terms consisting of $m_{id}$ and $p_{id}$ as the public part; a $k_{id}$, freshness *n*, and new short-term key *k* as the private part, protected by the Proto-LTK key *K*; and a digital signature computed over both the private and public part. After obtaining the terms, each $ECU_r^m$ is required to provide the proof of usage for the new key as a response.

If one protocol, either Proto-LTK or Proto-STK, fails, SKS offers the possibility of synchronisation. This protocol is an alternative to re-running the whole sequence from the beginning. SKS is designed to function for both Proto-LTK and Proto-STK, the differentiation between the two being accomplished via the protocol type term $p_{type}$.

$$ECU_i^m \rightarrow ECU_r^d : m_{id}, p_{id}\{p_{type}, n\}_{pk(r)}, \{m_{id}, p_{id}, \{p_{type}, n\}_{pk(r)}\}_{sk(i)}$$
$$ECU_r^d \rightarrow ECU_i^m : m_{id}+1, p_{id}\{n+1, p_{type}, k_{id}, K(i,r)\}_{pk(i)}\{m_{id}+1, p_{id},$$
$$\{n+1, p_{type}, k_{id}, K(i,r)\}_{pk(i)}\}_{sk(r)}$$
$$ECU_i^m \rightarrow ECU_r^d : m_{id}+2, p_{id}, \{n+2, , k_{id}\}_{K(i,r)}. \tag{3}$$

SKS is initiated by a protocol receiver $ECU_i^m$ by requesting via a challenge a new key to the distributor $ECU_r^d$. The challenge consists of a freshness nonce *n* and the $p_{type}$ signed with the private key of $ECU_i^m$. If successfully verified, $ECU_r^d$ responses with a new key depending on $p_{type}$. In Equation (3), the term *K* can denote both (a Proto-LTK or Proto-STK

key) in relation with $p_{type}$. Consequently, $ECU_r^d$ is required to confirm that the distribution of the new key was successful with an acknowledgment.

Additional KDS execution variants are addressed in Section 7 for completeness.

### 4.3. Data Authentication

Proposed in a previous work [18] as an alternative to the SecOC standard [13], Mix-CAN explored the idea of decoupling the data authentication process from the actual data transmission. This process implied frame aggregation over time, the computation of a predetermined number of MACs, and an aggregation process of the MAC tags using an Encrypted Bloom Filter (EBF). Consequently, the MixCAN scheme intended to reduce the bus overhead that can be brought by introducing new authentication messages.

From the point of view of security design, MixCAN was validated through a false-positive analysis manifested as a consequence of using EBF. In other words, there is a trade-off in the protocol in terms of security for performance. As MixCAN is a broadcast protocol without confirmation, the security analysis performed and presented later in the paper brings it to the surface that it lacks aliveness and agreement between the protocol participants. As such, a new extension of the protocol is offered to address those short-comings, where the properties are required. Therefore, the communication structure was improved with a confirmation message.

Let $ECU_i^s$ be the protocol initiator that periodically sends to a protocol receiver $ECU_r$ a sequence of frames $f_s$. After $x$ frames are sent, $ECU_i^s$ proceeds to computes $E_{bf}(f_s, n, k(i, r))$, where $E_{bf}$ is a function computing a new EBF, $n$ a shared freshness value, and $k$ a short-term authentication key obtained from Proto-STK. Once the EBF is computed, $ECU_i^s$ computes an additional authentication tag over it, and transmits the structure to $ECU_r$. If $ECU_r$ successfully verifies the obtained structure, it responds with a confirmation message to $ECU_i^s$. This procedure is outlined in Equation (4).

$$
\begin{aligned}
&ECU_i^s \rightarrow ECU_r : f_s \\
&ECU_i^s \rightarrow ECU_r : m_{id}, p_{id}, E_{bf}(f_s, n, k(i, r))\{m_{id}, p_{id}, E_{bf}(f_s, n, k(i, r))\}_{k(i,r)} \\
&ECU_r \rightarrow ECU_i^s : \{m_{id} + 1, n + 1, p_{id}\}_{k(i,r)}.
\end{aligned}
\tag{4}
$$

### 4.4. Firewall and Intrusion Detection System

In designing and developing the SF/IDS [20], the concepts of a CAN firewall and a rule-based IDS were investigated. The SF/IDS can be defined as a rule-processing engine capable of acting both as a SF and as an IDS. When configured to act as a SF, the engine monitors sequences of CAN frames based on their CAN identifier field in a stateful manner. Consequently, known perturbations in sequences of frames can be detected. In addition to this, an additional purpose of the SF is to enable CAN gateways, to restrict traffic coming from one network to another. The IDS configuration of the rule engine extends the functionality of the SF with the additional capabilities of performing deep-packet inspection at a CAN frame data level. This is accomplished in the same manner as in the SF, by leveraging a set of predefined rules with additional functionality that allows byte-wise logical operations.

While the initial design of the SF/IDS engine focused on detecting known attacks on CAN traffic, it did not consider the temporal aspect of frame transmission. In CAN systems, frames are exchanged following a periodic cycle, or when an event happens. Consequently, deviations from frame cycles can be a sign that an abnormality happened, such as a timing attack, unauthorised data transmission, fuzzy data, or denial of a service attack [25,31].

To address this problem, the IDS was extended with an additional transmission frequency rule table. The table defines the mean transmission cycle times for monitored frames, and a minimum and maximum accepted delay for intrusion detection.

*4.5. Secure Logging*

Logging and the later auditing play a critical role in the process of ensuring system security. The objective of the S-Log service [20] is to capture any security related events generated by other security services, and properly log and report them. In the present case, security events range from failed data authentication to failed key exchange and intrusions detected at the CAN bus level. Once a security event is generated, the S-Log is notified. The event is composed of the actual event message related to the event, signed with the TPM signature engine, using the S-Log's private key, and an associated hash-chain value. The S-Log chains together different security events using the TPM's PCR to link together events, and to maintain a tamper-proof relationship between them.

Once the S-Log completes these operations with the TPM, it publishes the security events to a corresponding service on the cloud. The V-Log service is responsible for validating the received events. When a new event is received, V-Log proceeds to verify it, by performing the same operation done by S-Log. If the digital signature and the hash-chain values are successfully verified, the matching event is marked as *verified*.

**5. Security Analysis**

The security analysis was performed using the Scyther formal language modeling tool developed by Cremers [32]. Similarly, for repeatability reasons, the protocols are described using the formal notation from [30,32]. Scyther allows protocol analysis under perfect encryption assumptions. Furthermore, Scyther leverages the Dolve–Yao [33] adversary model, which fits perfectly with the threat model from Section 5.1; since it assumes that the adversary possesses complete control over the communication channel, it can eavesdrop, manipulate, and replay network messages. In addition to this, the adversary is able to execute the same cryptographic protocols as the protocol actors if it has knowledge of the correct cryptographic keys.

A protocol $p(i,r)$ can be described as a sequence of messages exchanged between a protocol initiator $i$ and a responder $r$, denoted by $send(i,r,m) \in S$ and $recv(i,r,m) \in R$ events, where $m$ is a message, $S$ the set of send events, and $R$ the set of received events, respectively. Additionally, a given protocol $p(i,r)$ contains a sequence of terms $t \in T$, and a sequence of *claims* $c \in C$ events used for protocol evaluation [32]. A *claim* is an assurance for an agent that a certain property of the protocol holds. As such, a protocol can be described as:

$$p(i,r) = (\{T\}, [S, R, C]) \equiv \begin{cases} p(i) = (\{T_i\}, [S_i, R_i, C_i]) \\ p(r) = (\{T_r\}, [S_r, R_r, C_r]), \end{cases} \tag{5}$$

where $p(i,r)$ is the generic protocol, and $p(x)$ is the protocol from the perspective of $x$.

Through a protocol verification tool, a protocol designer can write a protocol specification to verify that the protocol design guarantees certain requirements for the protocol participants. The claims considered in the current protocol analysis are offered by the Scyther tool, and were identified based on the security properties mentioned in Section 3.3. Furthermore, claims in formal protocol analysis aim to prove the correctness of the protocol specification. The *claims* assumed in the current analysis follow the definitions from [30]:

- *Secrecy (secret)*: A secrecy claim for a term $t \in T$ of an agent $i$ is true, if and only if the term $t$ never becomes known to the intruder.
- *Aliveness (alive)*: A protocol $p(i,r)$ satisfies the property of aliveness if and only if after an agent $i$ executes a round of $p(i,r)$, $i$ is sure that it communicates with a trusted agent $r$, and $r$ executed an event.
- *Weak agreement (weakagree)*: A protocol $p(i,r)$ satisfies the property of weak agreement if and only if an agent $i$ is sure that agent $r$ executed the correct role in the protocol.
- *Non-injective synchronization (nisynch)*: Represents a strong form of authentication in protocol $p(i,r)$. The claim is satisfied if all messages exchanged in $p(i,r)$ have been sent or received by $i$ or $r$.

- *Non-injective agreement (niagree)*: A protocol $p(i,r)$ satisfies the claim of non-injective agreement, if whenever $i$ completes a run of the protocol believing it communicates with $r$, then $r$ has run the protocol before believing to be communicating with $i$. Consequently, $i$ and $r$ must agree on the contents of the messages exchanged.

The protocol analysis consists of three steps. In the first step, the initially proposed protocols [18,19] are described in Scyther and individually analyzed. After identifying the claims that were not assumed by the protocols, step two and three are carried out in parallel. Step two consists of improving the protocols to satisfy all raised claims. As for step three, the improved protocol versions are put in a multi-protocol analysis. This happened as an iterative process since individual claims must hold in multi-protocol claims [30]. For each protocol, a formal description is provided, with the results available in Table 2.

**Table 2.** Initial protocol analysis results, with improved individual and multi-protocol analysis.

| Protocol | Claim | Term | Proof of Correctness | | |
|---|---|---|---|---|---|
| | | | **Initial Version** | **Improved Version** | **Multi-Protocol Analysis** |
| $LTK(i,r)$ | secret | $K(i,r)$ | Ok | Ok | Ok |
| | secret | $n$ | Fail | Ok | Ok |
| | secret | $n+1$ | Fail | Ok | Ok |
| | secret | $k_{id}$ | Fail | Ok | Ok |
| | alive | - | Fail for $i$, Ok for $r$ | Ok | Ok |
| | weakagree | - | Fail for $i$, Ok for $r$ | Ok | Ok |
| | niagree | - | Ok | Ok | Ok |
| | nisynch | - | Ok | Ok | Ok |
| $STK(i,r)$ | secret | $k(i,r)$ | Ok | Ok | Ok |
| | secret | $n$ | Fail | Ok | Ok |
| | secret | $n+1$ | N/A | Ok | Ok |
| | secret | $k_{id}$ | Fail | Ok | Ok |
| | alive | - | Fail for $i$, Ok for $r$ | Ok | Ok |
| | weakagree | - | Fail for $i$, Ok for $r$ | Ok | Ok |
| | niagree | - | Ok | Ok | Ok |
| | nisynch | - | Ok | Ok | Ok |
| $SKS(i,r)$ | secret | $p_{type}$ | N/A | Ok | Ok |
| | secret | $n$ | Fail | Ok | Ok |
| | secret | $n+1$ | Fail | Ok | Ok |
| | secret | $n+2$ | N/A | Ok | Ok |
| | secret | $k_{id}$ | Fail | Ok | Ok |
| | secret | $K(i,r)$ | Ok | Ok | Ok |
| | alive | - | Fail | Ok | Ok |
| | weakagree | - | Fail | Ok | Ok |
| | niagree | - | Fail | Ok | Ok |
| | nisynch | - | Fail | Ok | Ok |

**Table 2.** *Cont.*

| Protocol | Claim | Term | Proof of Correctness | | |
|---|---|---|---|---|---|
| | | | **Initial Version** | **Improved Version** | **Multi-Protocol Analysis** |
| | secret | $n$ | Ok | Ok | Ok |
| | secret | $n+1$ | N/A | Ok | Ok |
| | secret | $E_{bf}$ | Ok | Ok | Ok |
| $MixCAN(i,r)$ | alive | - | Fail for $i$, Ok for $r$ | Ok | Ok |
| | weakagree | - | Fail for $i$, Ok for $r$ | Ok | Ok |
| | niagree | - | Ok | Ok | Ok |
| | nisynch | - | Ok | Ok | Ok |

*5.1. Threat Model*

The threat model considered in the current work assumes that a malicious actor possess capabilities in terms of (i) remote/physical access, (ii) system knowledge, and (iii) security protocol knowledge.

Let $a$ denote a protocol adversary. To attack a protocol $p(i,r)$, the considered threat model assumes that $a$ has physical access to the vehicle with or without the owner's consent (point (i)). Consequently, the adversary $a$ is capable of physically connecting to the network communication medium, replicating the network traffic of a legitimate protocol member $i$ (e.g., CCU) or $r$ (e.g., ECU) for an extended period of time. On the other hand, an attacker may compromise a protocol initiator $i$ (e.g., CCU in case of KDS), to obtain access to the underlying vehicle network. Once an attacker obtains an entry point to the vehicle network, it is assumed that it has knowledge regarding the protocol communications. Adversary $a$ can obtain knowledge about public protocol terms, such as $p_{id}$, $m_{id}$, $f_s$, or $f_r$. In other words, $a$ is able to eavesdrop on the network and record, modify, and replay network messages without generating communication errors (point (ii)). If $a$ does not compromise $r$ or $i$, thus obtaining access to $r$ (e.g., an ECU, or $i$ a CCU), authentication, confidentiality, and non-repudiation is kept, since $k$, $K$, and $sk$ or $pk$ are not known to $a$. Lastly, for point (iii) $a$ is able to execute the same cryptographic operations as $i$ and $r$. It is capable of obtaining knowledge regarding the running security protocols and further replicate them.

*5.2. Individual Service Analysis*

5.2.1. Long-Term Key Exchange

In the protocol $LTK(i,r)$, the role of the protocol initiator $i$ is assigned to the services responsible for generating a new key $K(i,r)$. In consequence, the responder $r$ roles are assigned to services listening and waiting for a new long-term symmetric key $K(i,r)$. As can be observed in Table 2, the initial protocol was expanded with a additional term $m_{id}$ for better message identification.

To satisfy the claims for aliveness and weak agreement for both $i$ and $r$, an additional message was introduced for $r$ to give $i$ proof that the new key $K(i,r)$ was received successfully, and that $r$ can perform an action with it. The improved $LTK(i,r)$ protocol satisfies all the claims previously mentioned, both in the individual and in the multi-protocol analysis. The formal description of the the $LTK(i,r)$ protocol is given in Equation (6).

$$LTK(i,r) = (\{i, r, m_{id}, p_{id}, k_{id}, n, K(i,r), pk(i), sk(i), pk(r), sk(r)\},$$
$$[send_1(i, r, m_{id}, p_{id}, \{k_{id}, n, K(i,r)\}_{pk(r)}, \{m_{id}, p_{id}, \{k_{id}, n, K(i,r)\}_{pk(r)}\}_{sk(i)})$$
$$recv_1(i, r, m_{id}, p_{id}, \{k_{id}, n, K(i,r)\}_{pk(r)}, \{m_{id}, p_{id}, \{k_{id}, n, K(i,r)\}_{pk(r)}\}_{sk(i)})$$
$$send_2(r, i, m_{id} + 1, p_{id}, \{m_{id} + 1, p_{id}, k_{id}, n + 1, \}_{K(i,r)})$$
$$recv_2(r, i, m_{id} + 1, p_{id}, \{m_{id} + 1, p_{id}, k_{id}, n + 1\}_{K(i,r)})$$
$$claim(i, secret, K(i,r), n, n + 1, k_{id})$$
$$claim(i, alive)$$
$$claim(i, weakagree)$$
$$claim(i, niagree)$$
$$claim(i, nisynch)]). \tag{6}$$

### 5.2.2. Short-Term Key Exchange

The $STK(i,r)$ protocol follows the same approach in terms of roles as $LTK(i,r)$. The $STK(i,r)$ initiator $i$ is the same service as in $LTK(i,r)$, and the responders $r$ are the equivalents in $LTK(i,r)$. The $STK(i,r)$ protocol suffered from the same security problems as $LTK(i,r)$. Therefore, a message identifier $m_{id}$ term was added and a proof message was introduced for $r$ to prove that the new symmetric short-term key $k(i,r)$ was successfully received to achieve aliveness and weak agreement. The protocol description is outlined in Equation (7).

$$STK(i,r) = (\{i, r, m_{id}, p_{id}, k_{id}, n, K(i,r), sk(i), k(i,r)\},$$
$$[send_1(i, r, m_{id}, p_{id}, \{k_{id}, n, k(i,r)\}_{K(i,r)}, \{m_{id}, p_{id}, \{k_{id}, n, k(i,r)\}_{K(i,r)}\}_{sk(i)})$$
$$recv_1(i, r, m_{id}, p_{id}, \{k_{id}, n, k(i,r)\}_{K(i,r)}, \{m_{id}, p_{id}, \{k_{id}, n, k(i,r)\}_{K(i,r)}\}_{sk(i)})$$
$$send_2(r, i, m_{id} + 1, p_{id}, \{m_{id} + 1, n + 1, p_{id}, k_{id}\}_{k(r,i)})$$
$$recv_2(r, i, m_{id} + 1, p_{id}, \{m_{id} + 1, n + 1, p_{id}, k_{id}\}_{k(r,i)})$$
$$claim(i, secret, k(i,r), n, n + 1, k_{id})$$
$$claim(i, weakagree)$$
$$claim(i, niagree)$$
$$claim(i, nisynch)]). \tag{7}$$

### 5.2.3. Symmetric Key Synchronisation

Compared to the two previously mentioned protocols, the roles in the $SKS(i,r)$ are reversed. The communication initiator $i$ requests from the responder $r$ a new symmetric key $K(i,r)$ depending on the protocol type $p_{type}$ term. Here, $K(i,r)$ is used interchangeably with $k(i,r)$, as the $SKS(i,r)$ protocol was designed to fit both long-term keys with $LTK(i,r)$ and short-term keys with $STK(i,r)$. Consequently, the value of $p_{type} \in \{LTK, STK\}$ is binary and must correspond to one of the protocols. The original design of the $SKS(i,r)$ followed a challenge–response approach.As the initial protocol analysis points out (see Table 2), while the secrecy of the new symmetric key is proved, the challenge containing the nonce $n$ is not protected. To address the unmet claims, the $SKS(i,r)$ protocol was extended in the same manner as the other two. The initiator $i$ requests a new key by challenging $r$, $r$ responds to the request, and finally, $i$ proves $r$ (that it can successfully use the new key by performing and event).

$$SKS(i,r) = (\{i, r, m_{id}, p_{id}, p_{type}, k_{id}, n, K(i,r), pk(r), sk(i)\},$$
$$[send_1(i, r, \{m_{id}, p_{id}, p_{type}, n\}_{pk(r)})$$
$$recv_1(i, r, \{m_{id}, p_{id}, p_{type}, n\}_{pk(r)})$$
$$send_2(r, i, \{m_{id} + 1, n + 1, p_{type}, p_{id}, k_{id}, K(i,r)\}_{pk(r)},$$
$$\{\{m_{id} + 1, n + 1, p_{type}, p_{id}, k_{id}, K(i,r)\}_{pk(r)}\}_{sk(i)}$$
$$recv_2(r, i, \{m_{id} + 1, n + 1, p_{type}, p_{id}, k_{id}, K(i,r)\}_{pk(r)},$$
$$\{\{m_{id} + 1, n + 1, p_{type}, p_{id}, k_{id}, K(i,r)\}_{pk(r)}\}_{sk(i)}$$
$$send_3(i, r, \{m_{id} + 1, n + 2, p_{id}, k_{id})\}_{K(i,r)})$$
$$recv_3(i, r, \{m_{id} + 2, n + 2, p_{id}, k_{id}\}_{K(i,r)})$$
$$claim(i, secret, p_{type}, n, n + 1, n + 2, K(i,r), k_{id})$$
$$claim(i, alive)$$
$$claim(i, weakagree)$$
$$claim(i, niagree)$$
$$claim(i, nisynch)]). \tag{8}$$

#### 5.2.4. MixCAN Protocol

The early concept for the data authentication protocol $MixCAN(i,r)$ decouples the actual message transmission of frames $f_s$ from the data authentication phase to reduce the amount of tags computed. For that reason, the authentication tag aggregation using the $E_{bf}$ function was performed after sending $f_s$, and finally sent without requiring a confirmation message. In addition to this aspect, the flaws in $MixCAN(i,r)$ are related to message numbering and protocol identification. In the present design, these points are addressed as can be seen in Table 2.

$$MixCAN(i,r) = (\{i, r, m_{id}, p_{id}, f_s, n, k(i,r), E_{bf}\},$$
$$[send_1(i, r, f_s)$$
$$recv_1(i, r, f_s)$$
$$send_2(i, r, m_{id}, p_{id}, E_{bf}(f_s, n, k(i,r)), \{m_{id}, p_{id}, E_{bf}(f_s, n, k(i,r))\}_{k(i,r)})$$
$$recv_2(i, r, m_{id}, p_{id}, E_{bf}(f_s, n, k(i,r))\{m_{id}, p_{id}, E_{bf}(f_s, n, k(i,r))\}_{k(i,r)})$$
$$send_3(r, i, \{m_{id} + 1, n + 1, p_{id}\}_{k(i,r)})$$
$$recv_3(r, i, \{m_{id} + 1, n + 1, p_{id}\}_{k(i,r)})$$
$$claim(i, secret, n, n + 1, E_{bf})$$
$$claim(i, niagree)$$
$$claim(i, nisynch)$$
$$claim(i, alive)$$
$$claim(i, weakagree)]). \tag{9}$$

#### 5.3. Multi-Protocol Analysis

Scyther provides an easy method for multi-service analysis. The only requirement is to concatenate the individual protocols subjected to analysis, and then run the analysis normally. All the above individually analyzed protocols were part of the multi-protocol analysis. The first multi-protocol analysis conducted pointed out a conflict between $LTK(i,r)$ and $SKS(i,r)$ due to message similarity. The motivation behind this was that $SKS(i,r)$ is an extension of $LTK(i,r)$ and $STK(i,r)$ to allow symmetric key receivers to request a new key in cases of error. As such, the protocol type term $p_{type}$ was introduced in $SKS(i,r)$ to

differentiate it from $LTK(i, r)$ and $STK(i, r)$. With this, message similarity was eliminated and the claims were proved. The results point out that every claim raised was proved to be correct, as can be seen in Table 2.

### 5.4. Threat Evaluation

To demonstrate the resistance against attacks of the improved protocol versions together with their possible drawbacks three common automotive threats, which were identified in [34], are presented. The main intention here is to link the claims considered in the formal analysis with concrete scenarios. For each attack, its corresponding attack scenario is presented in relation to our threat model, along with the countermeasures provided by the security services. As well as describing the resistance of the protocols against each attack, recommendations are provided to strengthen the system security in the cases of more powerful attacks.

#### 5.4.1. Man-in-the-Middle Attack

The attacker connects a malicious device to the CAN bus between two legitimate control units (e.g., CCU and ECU), capturing, manipulating, and afterwards forwarding received messages from one direction to the other. The aim of the attacker in this scenario is to compromise the data authentication protocol MixCAN and the KDS.

In the case of MixCAN, while the attacker is able to read the public part of the protocol $f_s$, $m_{id}$, $p_{id}$, and $E_{bf}$, they do not possess knowledge about the authentication key $k$, which is distributed via Proto-STK (i.e., the *secrecy* and *alive* claims stands). Compromising the message authentication tag computed with $k$ implies a brute-force attack on the authentication algorithm itself (e.g., advanced encryption standard). Consequently, the resistance against attacks of MixCAN is first related to the authentication algorithm considered. Secondly, message freshness is guaranteed by the nonce $n$, which is monotonically incremented with each protocol round. The probability of compromising the protocol by attacking its freshness is strictly related to nonce $n$ size in bytes, and the freshness update protocol. The freshness management was not considered in the current work, since standards such as SecOC provide a solid freshness management protocol [13]. Since the $E_{bf}$ is a probabilistic data structure manifesting a false positive rate on item query, in the prior work [18] this aspect was investigated, offering reference numbers for each parameter. Authentication (*aliveness*) and protocol agreement (*weakagree*, *niagree*) are guaranteed through message authentication or digital signature. This is further enforced with terms such as $n$, which is meant to link messages with its monotonic increments.

The authentication key $k$ is protected in the Proto-STK by the long-term encryption key $K$ (i.e., *secrecy* claim stands); its non-repudiation is assured by a bootstrapped secret key $sk$ (i.e., *nisynch*, *weakagree* claims stand), and a freshness nonce $n$ (i.e., *alive* claim stands). While the same arguments can be made about the security of $n$ as above, $sk$ and $K$ are protected by the secure storage of the TPM. Lastly, the *weakagree* and *niagree* claims are additionally satisfied through message acknowledgments which contain proof that a new term (e.g., new key) was verified and an authenticated response was received back. In the construction of all the proposed protocols, secrecy is guaranteed by means of authentication and encryption.

Since the attacker is physically positioned between two legitimate protocol members, it is obvious that any intervention on the bus introduces a certain delay. This delay is further increased when an attacker has to compute cryptographic operations. In this case, the SF/IDS possess the ability to monitor the transmission frequency of a set of messages and detect deviations or delays based on the maximum and minimum expected delays. Consequently, this limits the time window in which the attacker can read the messages from the network, conduct the attack, and replay the tampered messages.

### 5.4.2. Fuzzy and Replay Attack

While fuzzy attacks intend to perturb the normal functioning behavior of the underlying automotive system by sending messages containing arbitrary data, replay attacks assume that an attacker eavesdrops, reads, and replays back on the network previously observed messages with or without manipulation. Since the network communication is bus oriented, the process of conducting a fuzzy or replay attack manifests as a consequence the same outcome as the man-in-the-middle attack. In other words, these attacks are just a variation of the man-in-the-middle attack. Thus, the above scenario, claims, and countermeasures stand for these two attacks too. Even in more severe cases, where the attacker performs a more aggressive replay attack (e.g., denial of service), the SF/IDS come to aid. The SF/IDS are capable of running in both blacklisting and whitelisting mode, allowing identification of unauthorized messages on the network. Likewise, replay attacks and even denial-of-service attacks affect message transmission cycle time, which will trigger the SF/IDS. In the end, a trace of the security events generated by the SF/IDS will remain in logs generated by the S-Log service.

### 5.4.3. Compromise Attack

A more intelligent attacker may try to compromise a trusted protocol member (e.g., CCU or ECU). If the attacker is only able to perform read operations, the secrecy of keys is guaranteed by the TPM storage. On the other hand, if the attacker is also able to write (e.g., execute arbitrary code) he may gain access to the secret keys and run protocol rounds like a genuine protocol member. The severity of the attack depends on the compromised control unit (e.g., ECU or CCU). To resist against a compromised ECU, the proposed system requires additional grouping. Having multiple groups, each running a different instance of the KDS and MixCAN, is proven possible through the multi-protocol analysis and protocol terms (e.g., protocol identification). Consequently, the number of keys used grows proportionally with the number of groups.

If the attacker compromises a CCU, the most plausible counter-attack would be not be have a single control unit running multiple instances of the same protocol, but to have a designated ECU handling these processes inside of a specific group. Ultimately, the TPM standard allows security engineers to implement a secure and trusted boot on top of it which aims to protect the system against this type of attack. Consequently, modification to system code and parameters should be detected on boot.

## 6. Reference TestBed

The RTB intends to replicate a simple CAN system and to provide an easy-to-assemble system for testing security services targeting CAN systems. The reference architecture is built to be extensible in terms of hardware and integration of new services. New control units can be easily connected to the CAN network due to its bus nature. The same aspects were considered for integrating or replacing existing software services. Consequently, this section discusses the hardware used, the integration process of new software services, and interfacing with the existing services. Lastly, two distinct workflows are presented to showcase the functionalities of the RTB.

### 6.1. Reference Hardware

The hardware components used in the RTB were selected according to several criteria: accessibility in price, familiarity to the general user, and interoperability with drivers and other components. Figure 3 depicts the hardware setup used in the RTB. The main development boards are Raspberry Pi, mainly model 3B and 4, running Raspberry Pi OS Lite, with kernel version 5.15 and Debian bullseye. Each board has connected an OPTIGA SLx 9670 TPM 2.0, and a MCP2515 CAN controller with a TJA1050 CAN transceiver.

This configuration is not mandatory from the point of view of CAN communication, as can be seen in Figure 3. A third board can be seen with a SEEED CAN HAT daughter board. The motivation for the chosen configuration was to allow a physical TPM to be

connected to a Raspberry Pi together with the CAN controller. Alternatively, if this is not possible, the RTB can function normally with alternative virtual TPMs (e.g., IBM Virtual TPM (https://sourceforge.net/projects/ibmswtpm2/, access on 16 January 2023)). Wiring and additional details can be found on in Appendix A in Figure A1.

The OPTIGA SLx 9670 TPM 2.0 was considered to be adequate for the case at hand since it was developed by Infineon specially for automotive environments [35]. Additionally, Infineon recently published an automotive application guide [36] where they recommend the TPM for use by security applications, particularly for telematics units (e.g., CCU). While Infineon recommends that ECUs should possess hardware security modules, and considering the limitations of the RTB, a TPM was used also in the case of a ECU to emphasize service functionalities. In the end, the scope of the RTB is to showcase security features that can benefit the automotive security field.

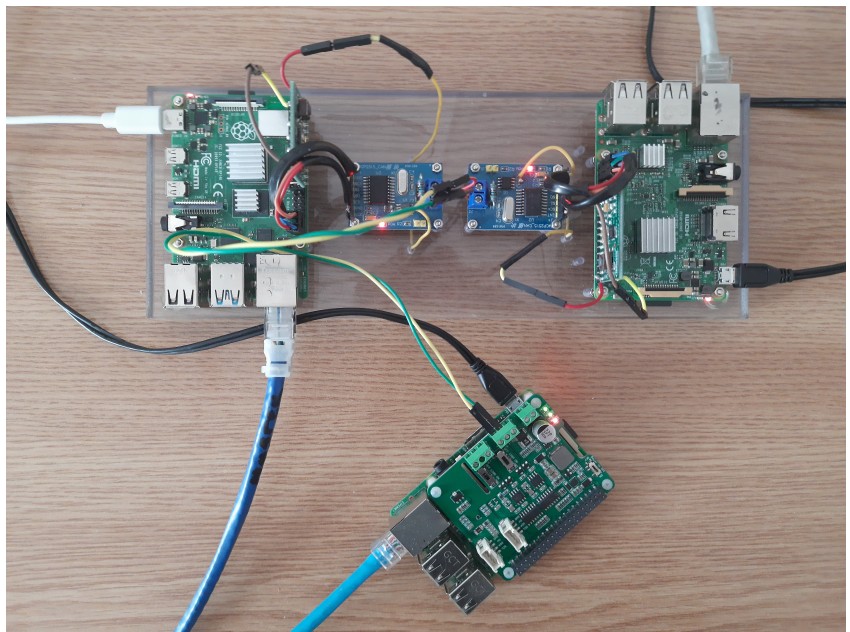

**Figure 3.** Reference testbed hardware setup incorporating Raspberry Pi boards, CAN controllers, and TPM modules.

### 6.2. Testbed Interfaces

To integrate a new service or to replace an existing one, there are three interfaces that must be taken in consideration: the CAN interface, internal Message Queue Telemetry Transport (MQTT) communication interface, and TPM interface.

#### 6.2.1. CAN Interface

The RTB allows a bi-directional CAN communication over a single CAN interface. While multiple services can connect to a CAN interface in read-only mode (e.g., MixCAN, SF/IDS), concurrent write operations are not allowed. Consequently, when multiple writing services are configured on a RTB instance (e.g., on a Raspberry Pi), a CAN bridge is considered. This connection consists of an internal virtual CAN bus that allows concurrent reads/writes, the actual CAN interface, and a CAN gateway. Using can-utils (https://github.com/linux-can/can-utils, accessed on 16 January 2023), the CAN gateway connects the CAN interface to the virtual one, forwarding messages coming from one to the other. Consequently, any messages sent on the virtual CAN are forwarded to the physical one, and vice versa.

#### 6.2.2. MQTT Interface

Depending on the scope of the new service, interfacing with the internal MQTT communication represents a must. This messaging approach also allows easy replacement of

the security services, such as the data authentication protocol MixCAN. If the new protocol uses symmetric cryptography, it is required to subscribe to the Proto-STK associated topic to periodically obtain a new symmetric key. This approach reduces the number of operations needed to parse the new key messages, compared to the alternative that implies listening on the CAN.

Likewise, the KDS can be replaced without difficulty if the new solution interfaces with the existing security modules (e.g., MixCAN) in terms of MQTT topics and message structure. New independent modules can be added and linked to the MQTT broker, with their own topics and messaging. This requires topic definition and configuration of the MQTT broker to authenticate the new module's users.

### 6.2.3. TPM Interface

The TPM is rich in features with a properly defined interface. The RTB interfaces with the TPM using the TCG TSS (https://github.com/tpm2-software, accessed on 16 January 2023). The TSS libraries offer a wide variety of functionalities, ranging from low level TPM commands to high level function calls to abstract the internal implementation. Additionally, using the TPM Access Broker & Resource Manager, the RTB is capable of functioning according to the service configuration, with physical connected TPM, or with a virtual one (e.g., IBM Virtual TPM). The TPM interface is language agnostic, and it can properly function with any library that respects the TSS specifications.

### 6.3. Workflow

To showcase the interaction between different security services, two scenarios were chosen for demonstration:

- Scenario 1: Showcases the process of distributing a new short-term authentication key using Proto-STK. Along with this, the MixCAN protocol is executed to present how it utilizes the key obtained from Proto-STK.
- Scenario 2: Outlines the life-cycle of a security event generated by the SF/IDS, how it is processed by the S-Log together with the TPM, and how it is published afterwards, and finally verified on the cloud side.

Each scenario is described using a sequence diagram. Each step in a diagram consists of one or more sub-steps.

### 6.3.1. Scenario 1

Outlined in Figure 4, the first scenario assumes that Proto-LTK was executed beforehand and consists of two phases. In the first phase, Proto-STK is executed, and in the second one the MixCAN protocol is carried out.

In phase one, the first step consists of a round of Proto-STK. The protocol initiator performs the necessary operations with the TPM to obtain a new authentication key. In step 2, the new key is encrypted with a Proto-LTK private key, and the necessary terms are used to construct the Proto-STK data structure. Step 3 converts the data structure obtained into several CAN frames and broadcasts it over the CAN bus. Finally, a protocol receiver obtains the Proto-STK CAN frames in step 4, where it proceeds to verify the information received and decrypt the new key using the shared long-term encryption key. Once the new key is received internally by MixCAN, the second phase begins.

Step 1 in MixCAN implies that CAN frames are exchanged normally, and periodically for each sent frame a MAC is computed on the sender with the Proto-STK key in step 2. Once a predefined number of MACs is computed, they are inserted into the EBF in step 3. Finally, an additional MAC is computed over the EBF in step 4, just to be sent in step 5. Similarly, on the receiver side, this process is repeated to verify the frames and authentication tags obtained.

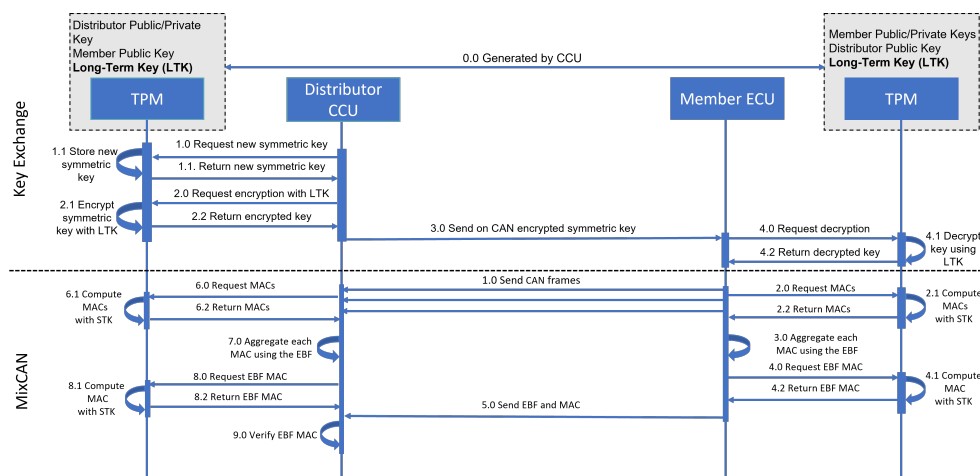

**Figure 4.** Workflow of distributing a new authentication key using the Proto-STK key exchange protocol and its usage by the data authentication protocol MixCAN.

### 6.3.2. Scenario 2

Figure 5 showcases the workflow of security events in the system. The sequence begins with the SF/IDS generating a security event once it detects abnormal behaviour on the CAN bus. This event is received by the S-Log, which proceeds to communicate with the TPM to construct the digital signature and extend its associated hash-chain in Step 1. Once this is accomplished, in the second step the information is published to the cloud. On the other side, in step 3 the V-Log periodically communicates with the cloud to check if new security events were pushed to the cloud database. If new entries are found, the V-Log requests them, and proceeds to verify them using its TPM and the S-Log public key in step 4. Finally, the status corresponding to the processed entry is updated in the database as verified or not verified.

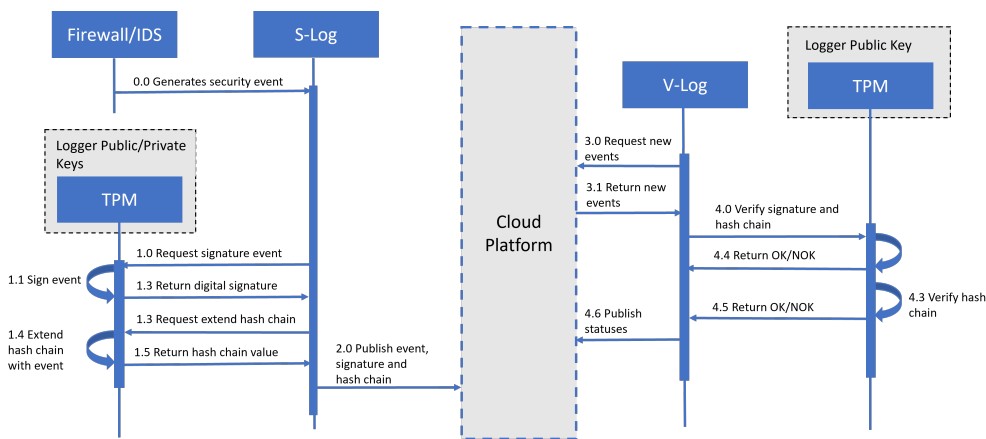

**Figure 5.** Workflow presenting the processing steps performed by the S-Log and V-Log services to sign and verify with the TPM a security event generated by the SF/IDS.

### 6.4. Experimental Assessment

The experimental assessment focuses on two aspects. First, it provides concrete performance measurements on time execution for the operations executed with the TPM on the reference hardware presented in Section 6.1. Secondly, it showcases 11 replay attacks conducted on the RTB to verify the message frame frequency monitoring introduced in the SF/IDS.

### 6.4.1. Performance Measurements

Several proposed security services (e.g., KDS, MixCAN, and S-Log) leverage the TPM for cryptographic operations and key storage. For a total of 10 TPM commands,

100 measurements were computed for each command, with the minimum, maximum, and average execution time. Table 3 presents the mentioned measurements, and provides a description for each command and its usage in the corresponding security services.

In Figure 6 the performance measurements for generating cryptographic keys and the key loading command are displayed. For example, the tpm2_create command has an average execution time of 10.237 s, with a maximum of 32.330 s, and a minimum of 0.806 s. To load the generated asymmetric key generated by tpm2_create, tpm2_load shows an average time of 0.441 s, a maximum of 0.425 s, and a minimum of 0.469 s. On the other hand, in Figure 7a, it can be observed that the tpm2_rsaencrypt command is faster than tpm2_rsadecrypt, with an average execution time of 0.168 s, while the latter time is 0.486 s. Similar numbers can be observed for the tpm2_verifysignature and tpm2_sign commands. Lastly, Figure 7b showcases the execution times for tpm2_pcrread and tpm2_pcrextend commands used by S-Log and V-Log.

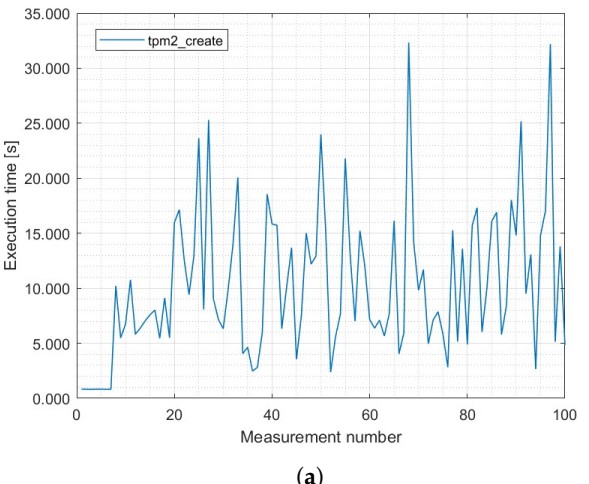
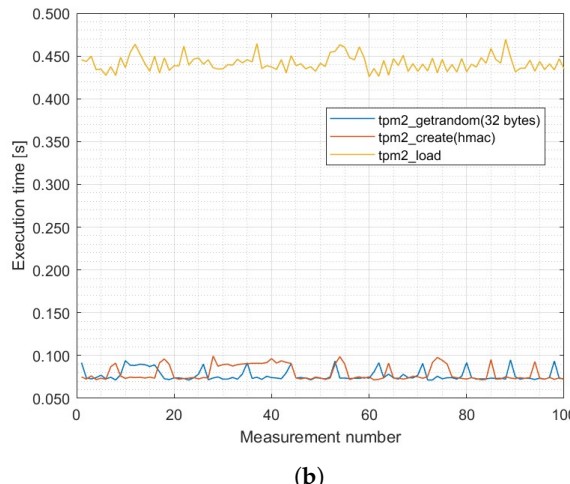

(**a**) (**b**)

**Figure 6.** Performance measurements of TPM commands that handle the generation of cryptographic keys and the loading process into the TPM. (**a**) tpm2_create command. (**b**) tpm2_getrandom, tpm2_create(hmac), and tpm2_load commands.

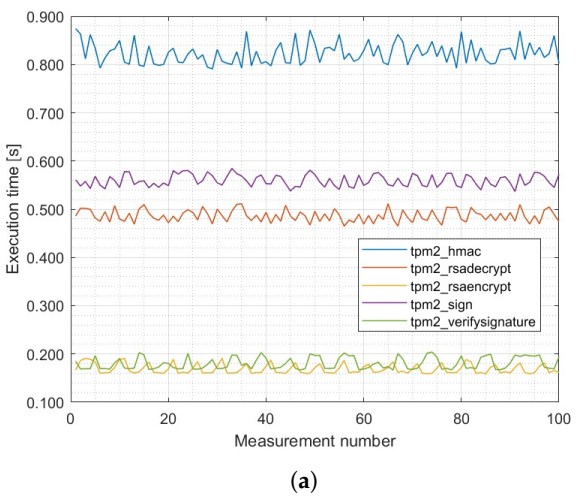
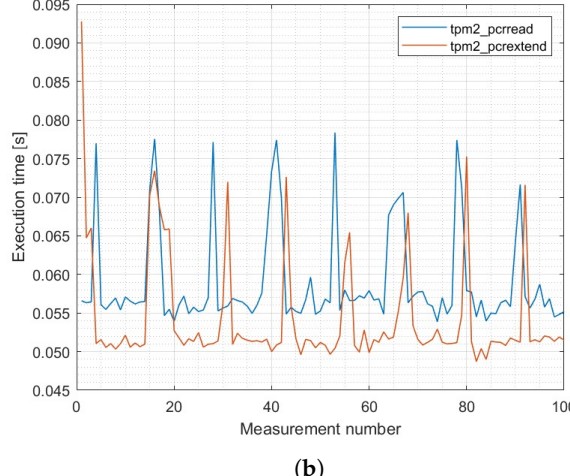

(**a**) (**b**)

**Figure 7.** Performance measurements of TPM commands that handle the cryptographic operations. (**a**) tpm2_hmac, tpm2_rsadecrypt, tpm2_rsaencrypt, tpm2_sign, tpm2_verifysignature commands. (**b**) tpm2_pcrread and tpm2_pcrextend commands.

**Table 3.** Minimum (Min.), maximum (Max.) and average (Avg.) execution time in seconds of TPM commands.

| TPM Command | Description | Min. [s] | Max. [s] | Avg. [s] |
|---|---|---|---|---|
| tpm2_create | Used in the bootstrapping process of the KDS and S-Log to generate a pair of asymmetric keys. | 0.806 | 32.330 | 10.237 |
| tpm2_getrandom | Used by Proto-LTK to generate an encryption key. | 0.071 | 0.094 | 0.076 |
| tpm2_create(hmac) | Used by Proto-STK to generate an authentication key. | 0.071 | 0.099 | 0.079 |
| tpm2_load | Used by KDS and S-Log on start-up to load the bootstrapped asymmetric keys. | 0.425 | 0.469 | 0.441 |
| tpm2_hmac | Used by MixCAN to compute authentication tags. | 0.790 | 0.874 | 0.823 |
| tpm2_rsadecrypt | Used by KDS for decryption. | 0.464 | 0.511 | 0.486 |
| tpm2_rsaencrypt | Used by KDS for encryption. | 0.158 | 0.191 | 0.168 |
| tpm2_sign | Used for computing digital signature by KDS and S-Log. | 0.536 | 0.584 | 0.559 |
| tpm2_verifysignature | Used for verification of digital signature by KDS, S-Log and V-Log. | 0.166 | 0.203 | 0.180 |
| tpm2_pcrread | Used by S-Log and V-Log to read the state of a PCR index. | 0.053 | 0.078 | 0.059 |
| tpm2_pcrextend | Used by S-Log and V-Log to extend the state of a PCR index. | 0.048 | 0.092 | 0.054 |

It is to be noted that the performed measurements were carried out on the RTB hardware at an operating system level. The reader should take into consideration that the measurements do not reflect the behavior of the TPM in real automotive environments, but only on this particular hardware with the constraints imposed by the operating system.

6.4.2. Attack Evaluation

While in Section 5 the network-related security services were analyzed, and a set of attacks were defined and discussed, in several cases the SF/IDS was mentioned as a countermeasure. Prior work [20] only addressed the performance aspect of the SF/IDS. Consequently, since the SF/IDS was improved with the capability of monitoring the message transmission frequency, especially for messages that are transmitted based on a cycle time, 11 variations of possible replayed attacks were conducted against the SF/IDS to determine its correctness. The first step in conducting these attacks consisted of constructing a trace log baseline. This process included a recording step, where a 30 s clean log file was obtained. The baseline log trace was split into three distinct parts for repeatability. Part 1 of the baseline trace consists of 10 s with attack-free messages, Part 2 the next 10 s representing the time frame where the attack will be carried out, and lastly, Part 3 consists of 10 s with attack-free messages. The first two attacks, as can be seen in Table 4, are focused on replaying a whole part of the baseline trace file in Part-2. On the other hand, the rest of the attacks are meant to modify the transmission and cycle time of a specific CAN frame. For example, for the CAN frame which was manipulated in the attacks with a mean cycle time of 30 ms, a minimum allowed delay of $\mu - 3\sigma$ ms was chosen, with the maximum of $\mu + 3\sigma$ ms.

**Table 4.** Attacks detected by the SF/IDS based on frame frequency monitoring. Each attack contains a 30 s pre-recorded trace file with three parts: Part 1 (10 s, no attack), Part 2 (10 s, with attack) and Part 3 (10 s, no attack).

| Attack Name | Attack Description | Detected by SF/IDS |
|---|---|---|
| Part 1 replayed | Part 1 replayed in Part 2 | ✓ |
| Part 3 replayed | Part 3 replayed in Part 2 | ✓ |
| Remove every 2nd frame | Remove every 2nd frame with CAN ID X in Part 2 | ✓ |
| Remove every 3rd frame | Remove every 3rd frame with CAN ID X in Part 2 | ✓ |
| Insert frames in Part 2 | Insert frame with CAN ID X between two of the same frames in Part 2 | ✓ |
| Delay every 2nd frame 10% | Delay every 2nd frame with CAN ID X by 10% of its cycle time in Part 2 | ✓ |
| Delay every 2nd frame 20% | Delay every 3rd frame with CAN ID X by 20% of its cycle time in Part 2 | ✓ |
| Increase cycle time 10% | Increase cycle time of every frame with CAN ID X by 10% in Part 2 | ✓ |
| Decrease cycle time 10% | Decrease cycle time of every frame with CAN ID X by 10% in Part 2 | ✓ |
| Shift forward every frame 20% | Shift forward by 10% the expected time slot of CAN ID X in Part 2. Cycle time is not modified. | ✓ |
| Shift backwards every frame 10% | Shift backwards by 10% the expected time slot of CAN ID X in Part 2. Cycle time is not modified. | ✓ |

*6.5. Repository*

All the developed software for the RTB is distributed under a MIT License and is publicly available (https://github.com/terilenard/can-tpm-reference-testbed, accessed on 16 January 2023).

**7. Discussions**

There are several points to be addressed regarding the proposed services: in particular, design considerations, computational impact, and their usability in a real world scenarios. For the initially proposed protocols, a higher concern was given to limit the number of cryptographic operations to reduce the overhead of the protocols introduced into the system. Consequently, there was a trade-off in terms of reducing the number of messages used in the protocols in opposition to security. In the present approach, a higher importance was given to having robust secure protocols with the possible disadvantage of having an increased overhead. Of course, this impact was supposed to be decreased by leveraging TPMs, separating in this way the cryptographic intensive operation from the main processing unit. At the same time, having additional confirmation messages and messages that prove certain actions were successfully executed (e.g., a key was distributed or data was authenticated successfully) represented a requirement from the point of view of the formal analysis, even if the communication is bus oriented.

While the SF/IDS and the S-Log are not concerned with the underlying communication and the data authentication protocol MixCAN has a low overhead, as demonstrated in the original work [18], the number of messages leveraged to distribute keys may increase CPU consumption and bus overhead. We propose to the system engineer to assign a CAN identifier for KDS protocol, such that all messages from one protocol are sent with the same priority over the bus. This will allow KDS to not interfere with high priority messages. Additionally, the protocol messages should be sent with a cycle time. This implies that each message from a protocol is sent at a known time cycle, giving more control to system engineers over the impact that KDS can have on the system.

Lastly, a point not addressed in the current work is log aggregation. Logging each incoming security event generates, in the end, difficulties in terms of storage and the amount of data that needs to be reported. Consequently, this represents an open problem that needs to be addressed.

## 8. Future Work

A key topic that was not addressed in the work at hand is the problem of the means of a trusted party to manage and interact with groups or fleets of vehicles. Long-term key updates for vehicle fleets in which TPMs are used represents a challenging task which is envisioned to be addressed by future work. For this investigation, the plan is to analyze centralized and decentralized approaches (e.g., public-key infrastructures, digital identities), identify the requirements of the OEM, which actors are involved in the whole process, and finally, offer an appropriate solution to fit the findings.

While the work at hand is strictly focused on the security aspect of in-vehicle systems, the system safety dimension requires attention. Prior works focused on identifying the key challenges in cybersecurity, privacy and standards of automated shuttles [37], and on the GDPR implications on data privacy targeting the same systems [38]. In parallel with the previous direction, our future work additionally intends to improve the threat analysis and risk assessment (TARA) method with the necessary processes to cover autonomous mobility systems requirements.

## 9. Conclusions

The main contribution of the work at hand represents automotive RTB empowerment by TPMs, incorporating multiple trusted security services. The RTB aims to replicate a simple CAN system designed for security-related experiments. In the RTB, long-term encryption keys and short-term authentication keys are periodically exchanged, data is sent authenticated, the network is monitored by a stateful firewall and intrusion detection system, and security events are logged with the TPM. As for the second contribution, the security services proposed were improved from previously published versions in terms of protocol aliveness, agreement, authentication of terms, protocol, and message identification. The improved protocols were validated through a formal individual and multi-protocol analysis using the Scyther modeling tool under the Dolve–Yao adversary model. The analysis formally proved the correctness of the protocols. Lastly, the usage of TPMs is demonstrated in the RTB as a means of service and control units identification, key generation, storage, distribution, and log verification.

**Author Contributions:** Conceptualization, T.L., B.G., P.H., A.C., and N.A.N.; methodology, T.L. and P.H.; software, T.L., B.G., and P.H.; validation, T.L., B.G., and P.H.; formal analysis, T.L.; writing—original draft preparation, T.L.; writing—review and editing, T.L., B.G., P.H., A.C., and N.A.N.; visualization, T.L.; supervision, B.G., P.H., A.C., and N.A.N. All authors have read and agreed to the published version of the manuscript.

**Funding:** This work has received funding from the Swiss State Secretariat for Education, Research and Innovation (SERI) and co-funded by the European Union (grant agreement No 101077587). Views and opinions expressed are however those of the author(s) only and do not necessarily reflect those of the European Union or CINEA. Neither the European Union nor the granting authority can be held responsible for them.

**Conflicts of Interest:** The authors declare no conflict of interest.

## Appendix A

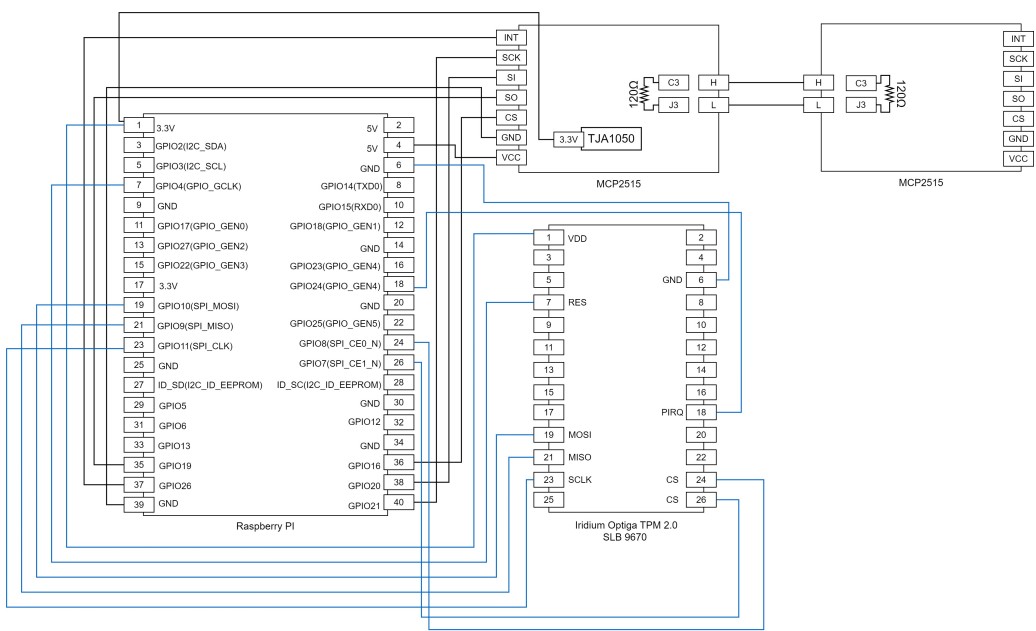

**Figure A1.** Input/Output schematic of the proposed testbed. In the schematic, the following components are showcased: a Raspberry Pi model 3B, MCP2515 CAN controller with TJA1050 CAN transceivers, and Iridium Optiga TPM 2.0 SLB 9670.

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
