# Peer review of "An Automotive Reference Testbed with Trusted Security Services"

_electronics, doi:10.3390/electronics12040888_

Round 1

Reviewer 1 Report

The paper deals with the security of automotive transport systems in the field of development of the specialized testbed with trusted security services. These days the automotive domain highly evolves considering new technologies (AI/ML, ADAS, Vehicle Dynamics, etc.) and internationally industry-approved norms and regulations of its development (AUTOSAR, ASPICE, etc.).

The paper can be of interest to developers and quality assurance engineers working on embedded systems in the automotive domain.

The paper is recommended for publication after the following comments and remarks:

-  It is advised to place the objects on the same page with the references. In addition, it is advised to enhance the titles of the objects (i.e. "Table 1. Table of symbols." - which symbols? ; "Figure 2. Proposed architecture." - ??? etc.). ;

- as safety is a top priority for such types of autonomous mobility systems (and not only to autonomous but let's say proven in-use "manual driving" systems) and currently there are international regulations on safety (i.e. ISO 26262) the interplay between security and safety is a challenging topic. The more secure system is (thus more complex by design) the less safe it would highly likely be (due to an increased amount of technical and architectural complexities which influence overall safety and can be used as a target of attacks). So here authors are advised to consider the possible influence of the proposed solution on the systems' safety. How it can be addressed?;

- The important aspect of each safety-critical system is licensing (certification, etc.). Here the question will be how and in which ways the proposed solution helps the abovementioned processes? Whether there are any issues?

- The paper is written for security specialists (security al)

- How the security claims have been identified and whether the used way can be trusted? To which extent? Whether the cross-check of claims has been done (in the sense of objectivity, impartiality, evidentiality, verifiability, justifiability, trustability, etc.)?

- What about a more precise (to my knowledge) requirement-based assessment in comparison to a claim-based assessment (where the claim is a more general description of a possessed characteristic and it needs not only further detalization but a special set of <property, evidence, action> to prove that a certain claim is implemented)?

It can be proposed (for future works of authors) to study the different notations of safety and security argumentation strategies and building safety cases, security cases (including ASAC,  Advanced Security Assurance Case) in order to represent the information about claims not only in a textual form (like now in the paper), but with a special notation (like GSN, Toulmin, CAE, ASAC notation, etc.). It can be useful for structuring the argument on safety/security and possible extention of the current work. Treat the second part of this comment as a friendly recommendation, but not as a reviewer's comment to be addressed directly in the current paper.

- Authors are advised to provide the schematic representation of the Testbed (including the I/O interconnections and buses) instead of a picture;

- Authors are advised to avoid empty fields on pages (see page 18);

- Authors provided a description of the security protocols in the paper. What can be interesting for readers - some more numbers (of the implementation of the protocols, its' run) on a particularly chosen hardware with some comparisons to other solutions in order to prove the trustability of the proposed solution.

Author Response

Dear reviewer,

Please find attached the cover letter containing the answer we provided to your feedback, together with the new version of the manuscript.

Best regards,

Teri Lenard.

Reviewer 2 Report

This paper proposes a Trusted Platform Module and develops a Reference Testbed for automotive systems. The proposed method and testbed are explained in detail and validated by the Scyther modeling tool. 

After reading the paper, I feel the authors could provide more argument or proof on how the Scyther modeling tool works and why the proposed method are robust against attacks and intrusions.  In the real life, there are many ways the security system is involved but the authors only showcase two workflow scenarios. More workflow scenarios would better convince the readers that the proposed method and testbed are robust again attacks and intrusions. 

In addition, there are some minor errors listed below.

RTB was defined in line 7, and therefore should not be defined again in line 8. 

In line 587, the first available should be removed. In line 611 and 612, in coming should be incoming. 

In line 18, Vehicless should be Vehicles. 

In line 129, crytographic should cryptographic. 

In line 181, intents should be intends.

In line 197, Identiy should Identity.

In line 320, denotes should be denote. 

In line 325, “a alternative” should be “an alternative”.

In line 328, “a aggregation” should be “an aggregation”.

In line 433, individuals should be individual.

In line 493, an should be deleted. 

In line 529, amount should be number. 

In line 540, the comma (,) should be deleted. 

In lines 555, 558 and 560, from should be of.

In line 582, it’s should be its. 

In lines 591, 593, 604, amount should be number. 

Author Response

Dear Reviewer,

Please find attached the cover letter with our responses to your feedback.

Best regards,

Teri Lenard.

Round 2

Reviewer 1 Report

The paper is recommended to be published. Good luck with ongoing research.